# Determinants of cigarette/bidi smoking among youth male in rural Mymensingh of Bangladesh: A cross-sectional study

**K. M. Mustafizur Rahman**[1], **Md. Ismail Tareque**[2]*

**1** Department of Population Science, Jatiya Kabi Kazi Nazrul Islam University, Mymensingh, Bangladesh,
**2** Department of Population Science and Human Resource Development, University of Rajshahi, Rajshahi, Bangladesh

* tarequemi_pops@ru.ac.bd, tareque_pshd@yahoo.com

**Data Availability Statement:** All relevant data are within the manuscript and Supporting Information files.

## Abstract

### Background

Smoking cigarette/bidi, is a serious health threat, causes preventable premature morbidity and mortality. Higher prevalence of smoking among the youth hampers a country's development, as the youth are the main drivers of socio-economic development. An effective understanding of factors associated with youth smoking is precious to prevent youth smoking. This study aims to identify the determinants of smoking cigarette/bidi among the youth male of the rural areas of Mymensingh district in Bangladesh.

### Methods

The primary data from the project "Knowledge, awareness and practices among youth smokers in Trishal Upazila under Mymensingh district: A micro-survey study", funded by the Research and Extension Center, Jatiya Kabi Kazi Nazrul Islam University, Bangladesh was utilized in the current study. The data consists of 385 youth males aged 15–24 years who were interviewed face-to-face from the rural areas of Mymensingh district in Bangladesh. Univariate distribution, chi-square tests, and binary logistic regression model were employed to identify the factors associated with smoking cigarette/bidi among the youth male.

### Results

The prevalence of smoking cigarette/bidi among the youth male is 40.3% [95% CI: 35.0%-45.0%]. Age, occupation, monthly income, family's monthly income, cigarette/bidi smoking status of father, brother and close friends, and knowledge about harmfulness of smoking are revealed as the determinants of cigarette/bidi smoking. For instance, the odds of being smoker increases with the increase in age (Odds ratio [OR]: 1.33 [1.17–1.51]). Business owner is less likely (OR: 0.15 [0.03–0.68]) to smoke than the day labourer. Having smoker fathers (OR: 2.51 [1.39–4.53]), smoker brothers (OR: 2.88 [1.39–5.96]), smoker friends (OR: 9.85 [5.85–1.27]) are significantly associated with smoking cigarette/bidi.

**Funding:** The author(s) received no specific funding for this work.

**Competing interests:** The authors have declared that no competing interests exist.

## Conclusion

As the first study, it provides the determinants of cigarette/bidi smoking among youth male of the rural areas of Mymensingh district in Bangladesh. Relevant authorities are suggested to consider the study's findings and recommendations to revise the existing smoking policies so that smoking among youth can be prevented for future development of the country.

## Introduction

Tobacco use is a significant public health concern. Globally, more than 7 million people die due to direct tobacco use, and around 1.2 million non-smokers die due to being exposed to second-hand smoke each year [1]. All forms of tobacco are harmful, and there is no safe level of exposure to tobacco [1]. Cigarette/bidi smoking is the most common form of tobacco use among men in Bangladesh, where 43% of adults use some form of tobacco, 45% of men and only 1.5% of women smoke cigarettes [2]. Despite having ban on the sales of tobacco to and by minors, particularly those aged <18 years, in Bangladesh national tobacco control law, university male students are found to initiate smoking at an average age of 17.8 years [3]. Evidences demonstrate that majority of the people try their first cigarette before the age of 18 and become daily smokers in adolescents. For instance, most of the European people start smoking before the age of 18 and often transit to regular smokers during young adulthood [4]. In the United States, about 90% of all regular smokers begin smoking at or before age 18, and hardly any regular smoker tries the first cigarette outside of childhood [5, 6]. When people start smoking at an earlier age, the duration of smoking must be longer for the people at adulthood. Due to the prolonged smoking, a wide range of diseases can accelerate the premature morbidity and mortality among the adults [7]. The longer duration of smoking creates more miserable conditions to their lives, families, societies and the country, especially for developing country like Bangladesh; because cigarette smoking harms almost every organ of human body, causes many diseases, and reduces the health of smokers in general [8]. Therefore, it is imperative to study the factors that are associated with smoking, particularly among the youth, in Bangladesh. Identification of modifiable factors will provide the possibilities to prevent smoking among the youth.

Bangladesh has made a considerable progress in socio-economic and demographic indicators over the last few decades [9]. The socio-economic and demographic indicators are progressing in the right path to achieve the desired level of prospective socio-economic development of the country. However, the progress could stay behind the desired level due to the use of tobacco among the youth, as the use of tobacco has high impact on growing economy and high expenditure on health [10]. Bangladesh is passing through its golden opportunity of demographic dividend where youth remains central to such development. Therefore, smoking among youth is an alarming issue to be taken care of for the future development of the country.

According to the Global Adult Tobacco Survey, in Bangladesh, around 22 million people are smoking different types of tobacco products, among them 12% people aged 15–24 years (considered youth in the current study; discussed in the Methods section) are consuming some forms of tobacco [11]. The number of tobacco users, particularly among the youth, is increasing day by day due to availability of cheap tobacco products, poor status of tobacco control regulations, and weak enforcement of such regulations [3, 12]. Youth is an important age group as healthy youth leads to healthy future. They are the main instruments for future development of the country. Higher incidence of smoking among the youth surely hampers the

future development. Hence, it is indispensable to restrict the youth from smoking. In different settings, factors associated with cigarette smoking among the youth includes exposure to smokers (elders, parents, friends, teachers), availability of tobacco, low socio-economic status, low self-esteem, poor academic performances [13, 14]. Although there are studies on the determinants of smoking around the world [15–20], in Bangladesh, extant literature has not paid particular focus on the youth section of the population. For example, Flora and others studied 35,446 individuals aged 18 and 90 years, and found an overall prevalence of smoking as 20.5% during 2001–2003 [21]. Using data from 12,155 men aged 15–59 years, Khan and others found the overall prevalence of smoking as 53.6%, with a higher prevalence in slum (59.8%) than non-slum (46.4%) in 2006 [22]. Sreeramareddy and others used nine south and south-east Asian countries' data sets from Demographic and Health Surveys, and found the prevalence of smoking as 60.0% among Bangladeshi men of age 15–54 years in 2007 [23]. They also found lower education and poverty to be responsible for smoking among Bangladeshi men. Using the Global Youth Tobacco Survey Bangladesh 2007 data, the prevalence of ever cigarette smoking was reported as 15.8% and the prevalence of smoking as 12.3% among boys who were 13–15 years of age and studying in grades 7–10 [24, 25]. Among the university male students aged 18–26 years, the prevalence of cigarette smoking was reported as 49.1% [3]. Due to lack of research on smoking among the youth aged 15–24 years in Bangladesh, we set the aim to identify the determinants of smoking among the youth in Bangladesh. It is believed that findings of the current study will be helpful for Bangladeshi policy makers in formulating strong measures along with revising or redrawing the existing policies to discourage smoking among the youth in Bangladesh.

## Methods

### Data

The primary data from the project "Knowledge, awareness and practices among youth smokers in Trishal Upazila under Mymensingh district: A micro-survey study", funded by the Research and Extension Center, Jatiya Kabi Kazi Nazrul Islam University, Bangladesh was utilized in the current study. In brief, following two inclusion criteria: (a) individuals of age 15–24 years and (b) male, the data for the project were collected from youth male aged 15–24 years (according to United Nations, individuals aged 15–24 years are youth [26]) from the rural areas of Trishal Upazila of Mymensingh district of Bangladesh from November 25, 2019 to December 15, 2019. The Upazila is about 20 kilometers away from the divisional town of Mymensingh, and about 90 kilometers away from the capital city of Dhaka. The Upazila consists of 1 municipality and 12 unions.

For a confidence level of 95% (Z = 1.96) and 50% prevalence of smoking among the youth (p = 0.50; 50% prevalence was considered due to unknown smoking prevalence among the youth), considering 5% margin of error (e = 0.05) with the formula

$$n = \frac{Z^2 p (1-p)}{e^2}$$

the minimum required sample size was 384~385. To reach required sample size, a multi-stage sampling technique was adopted. Using simple random sampling technique, at first one union (namely, Trishal Union) from Trishal Upazila, and then five wards (Ward 1, 2, 3, 8 and 9) from Trishal Union were selected. Finally, using systematic random sampling technique, data from 385 youth males were collected from the selected wards. Based on the total number of youth male in the selected wards, obtained from the Bangladesh Population and Housing Census 2011 [27], the data were collected, and during data collection one of every fifth male youth

was interviewed in the selected wards. Consequently, 53 youth males out of 265 from Ward 1, 81 out of 404 from Ward 2, 74 out of 370 from Ward 3, 91 out of 454 from Ward 8, and 86 out of 432 from Ward 9 were interviewed. A well-structured Bangla questionnaire containing demographic (age, number of family members etc.), socio-economic (education, occupation, monthly income etc.) and smoking related (smoking status, duration of smoking, smoking status of family members and friends etc.) questions was administered face-to-face to 385 youth males by well-trained interviewers. All respondents were asked to provide verbal consent after being read the consent statement from the questionnaire. The Bangla and English version of questionnaire are provided in the (S1 File and S2 File). From socio-cultural and religious points of view, smoking among female are still beyond the consideration in Bangladesh. Therefore, only males were considered for the project. The complete dataset is also provided in the (S3 File).

## Outcome variable

Current smoking status of the respondents (youth males aged 15–24 years) is considered as the outcome variable, which was dichotomized by assigning 1 if a respondent had been cigarette/bidi smoking for last six months prior to the survey, and 0 for otherwise.

## Explanatory variables

A wide range of individual and household characteristics, reported to be associated with smoking status among the youth in previous studies [20, 28–32], were included in the current study. Individual's characteristics include respondent's current age (classified as 15–17, 18–20, or 21–24 years; categorical age was considered for descriptive statistics and single year age for regression analysis), education (categorized 0 to 5 years of schooling as below secondary, 6 to 10 years of schooling as secondary, or 11+ years of schooling as higher secondary and above), occupation (categorized as day labourer, service, business, students, or unemployed), monthly income (no income, <5000 Bangladesh currency Taka [BDT], or ≥5000 BDT), financial dependency on the family (those having no income and depend totally on their family for any financial supports were classified as dependent, or otherwise independent), knowledge about the risks of disease– whether smoking brings the risk of chronic disease (no or yes), and knowledge about the harmfulness– whether smoking is harmful to other non-smokers (no or yes). Household's characteristics include education of father (categorized as below secondary, secondary, or higher secondary and above), education of mother (categorized as below secondary, secondary, or higher secondary and above), family income (<15000 BDT, 15000–25000 BDT, or >25000 BDT), current smoking status of father (no or yes), current smoking status of brothers (no or yes). Additionally, current smoking status of close friends (no or yes) was considered as explanatory variable, as youths are significantly influenced by their close friends [33]. The respondent's father, brother and close friends were considered as smokers if they had been cigarette/bidi smoking for last six months prior to the survey.

## Statistical analysis

After descriptive statistics of the study sample, chi-square tests were used to identify differences in the percentage of smokers by the explanatory variables (detailed above). All variables significant in chi-square tests at level p<0.20 were included in regression analysis [34]. Moreover, multicollinearity in the logistic regression analysis was checked by examining the standard errors for the regression coefficients. A standard error larger than 2.0 indicates numerical problems such as multicollinearity among the explanatory variables [35]. No evidence of multicollinearity was observed in the current study. Finally, a binary logistic regression model was

used to identify the factors that are influential in determining the smoking status of the respondents. The statistical significance of all analysis was set at p<0.05. No sampling weights were provided with the data as such we do not apply sampling weights in the analyses. The entire statistical analysis of the study was performed with SPSS version 16.0 for Windows (SPSS Inc., Chicago, IL, USA).

### Ethical considerations

The study design and funding for the project titled "Knowledge, awareness and practices among youth smokers in Trishal Upazila under Mymensingh district: A micro-survey study" were approved by the Research and Extension Center, Jatiya Kabi Kazi Nazrul Islam University (JKKNIU), Bangladesh. At the very beginning of the project questionnaire, a consent statement emphasizing voluntary participation in the study on knowledge, awareness and practices about smoking among youth assured that the respondents' information will be kept strictly confidential and the data will be utilized only for research purpose. The ethics committee of the Research and Extension Center, JKKNIU granted a waiver of ethical approval, as the project had no minimal risk to the subjects. All respondents were asked to provide verbal consent after being read the consent statement. They were not asked for written consent as some of them might have inability in reading and writing, which could make them reluctant to respond to the interview, and consequently the data collection process would have been jeopardized. Any identifying information was removed from the dataset.

## Results

Table 1 presents the socio-demographic characteristics of the study participants and the bivariate association between cigarette/bidi smoking status and socio-demographic characteristics to show the variation in percentage of smokers by socio-demographic characteristics. The average age of the participants is 18.9 years. Among the participants, 40% are 15–17 years, 28.8% are 18–20 years, and 31.2% are 21–24 years old. Average year of schooling is 9.6, and only 12.5% of respondents are with below secondary level of education. Respondents' fathers are with more years of education than their mothers (average year of education: father 7.3 years and mother 5.9 years). Majority of the respondents are students (67%). Although majority (66.8%) has no income, the average monthly income is BDT 4,439. The average monthly income of respondent's family is BDT 22,634, with around half (46%) of family's income in between BDT 15,000–25,000. 71.7% of respondents depend on their family for their financial supports while remaining 28.3% do not depend on their family for their financial supports. 57.4% of the respondents' fathers are currently cigarette/bidi smoking, 17.7% of respondents' brothers are currently cigarette/bidi smoking, and 36.1% of the respondents' close friends are currently cigarette/bidi smoking. Overwhelming majority of the respondents believe that smoking brings the risks of chronic diseases, and smoking is harmful to other non-smokers.

Among the respondents, 40.3% (95% confidence interval: 35.0%-45.0%) are currently cigarette/bidi smoking. On an average, the smokers are smoking 7.5 cigarette/bidi per day (Fig 1).

In Table 1, a significant association between cigarette/bidi smokers and age group was found; percentage of cigarette/bidi smokers increase with the increase in age (*p<0.001*). Respondents who are with below secondary education, day labourers, with monthly income BDT<5,000, with monthly family income BDT>25,000, financially independent from their family are more likely to smoke cigarette/bidi than their counterparts. Respondents whose fathers, brothers and close friends are cigarette/bidi smokers are also more likely to smoke cigarette/bidi than the respondents whose fathers, brothers and close friends are non-smokers.

**Table 1. Socio-demographic characteristics of the respondents, and percentage of smokers by socio-demographic characteristics (n = 385).**

| Variables | Frequency | Percentage | Smoker | |
|---|---|---|---|---|
| | | | Number | Row percentage |
| Age group | | | | |
| 15–17 | 154 | 40.0 | 30 | 19.5 |
| 18–20 | 111 | 28.8 | 45 | 40.5 |
| 21–24 | 120 | 31.2 | 80 | 66.7 |
| | | | | p-value = <0.001 |
| Average age (SD) | 18.9 (3.1) | | | |
| Education | | | | |
| Below secondary | 48 | 12.5 | 28 | 58.3 |
| Secondary | 197 | 51.1 | 53 | 26.9 |
| Higher secondary and above | 140 | 36.4 | 74 | 52.9 |
| | | | | p-value = <0.001 |
| Average year of schooling (SD) | 9.6 (3.3) | | | |
| Education of father | | | | |
| Below secondary | 169 | 43.8 | 65 | 38.5 |
| Secondary | 108 | 28.1 | 42 | 38.9 |
| Higher secondary and above | 108 | 28.1 | 48 | 44.4 |
| | | | | p-value = 0.57 |
| Average year of schooling of father (SD) | 7.3 (5.4) | | | |
| Education of mother | | | | |
| Below secondary | 207 | 53.8 | 77 | 37.2 |
| Secondary | 125 | 32.4 | 54 | 43.2 |
| Higher secondary and above | 53 | 13.8 | 24 | 45.3 |
| | | | | p-value = 0.40 |
| Average year of schooling of mother (SD) | 5.9 (4.7) | | | |
| Occupation | | | | |
| Day labourer | 20 | 5.2 | 16 | 80.0 |
| Service | 20 | 5.2 | 12 | 60.0 |
| Business | 65 | 16.9 | 34 | 52.3 |
| Students | 258 | 67.0 | 83 | 32.2 |
| Unemployed | 22 | 5.7 | 10 | 45.5 |
| | | | | p-value = <0.001 |
| Monthly income (in BDT) | | | | |
| No income | 257 | 66.8 | 79 | 30.7 |
| < 5000 | 21 | 5.5 | 13 | 61.9 |
| ≥5000 | 107 | 27.7 | 63 | 58.9 |
| | | | | p-value = <0.001 |
| Average income (SD) | 4439 (7877.8) | | | |
| Family's monthly income (in BDT) | | | | |
| < 15000 | 97 | 25.2 | 26 | 26.8 |
| 15000–25000 | 177 | 46.0 | 75 | 42.4 |
| > 25000 | 111 | 28.8 | 54 | 48.6 |
| | | | | p-value = <0.001 |
| Average monthly income of family (SD) | 22634 (12958.1) | | | |
| Financially dependent on the family | | | | |
| Dependent | 276 | 71.7 | 93 | 33.7 |
| Independent | 109 | 28.3 | 62 | 56.9 |

(*Continued*)

**Table 1.** (Continued)

| Variables | Frequency | Percentage | Smoker | |
|---|---|---|---|---|
| | | | **Number** | **Row percentage** |
| | | | **p-value = <0.001** | |
| Current smoking status of father* | | | | |
| No | 164 | 42.6 | 55 | 33.5 |
| Yes | 221 | 57.4 | 100 | 45.2 |
| | | | **p-value = <0.02** | |
| Current smoking status of brother | | | | |
| No | 317 | 82.3 | 114 | 36.0 |
| Yes | 68 | 17.7 | 41 | 60.3 |
| | | | **p-value = <0.001** | |
| Current smoking status of close friend | | | | |
| No | 246 | 63.9 | 54 | 22.0 |
| Yes | 139 | 36.1 | 101 | 72.7 |
| | | | **p-value = <0.001** | |
| Smoking brings the risks of chronic diseases | | | | |
| No | 6 | 1.6 | 5 | 83.3 |
| Yes | 379 | 98.4 | 150 | 38.8 |
| | | | **p-value = <0.03** | |
| Smoking is harmful to other non-smokers | | | | |
| No | 14 | 3.6 | 11 | 78.6 |
| Yes | 371 | 96.4 | 144 | 38.8 |
| | | | **p-value = <0.001** | |

**Notes:** BDT: Bangladesh currency–Taka; SD indicates Standard Deviation

* No mother of the study participants found as smoker; The p-values are of chi-square tests; P-values <0.20 are in boldface.

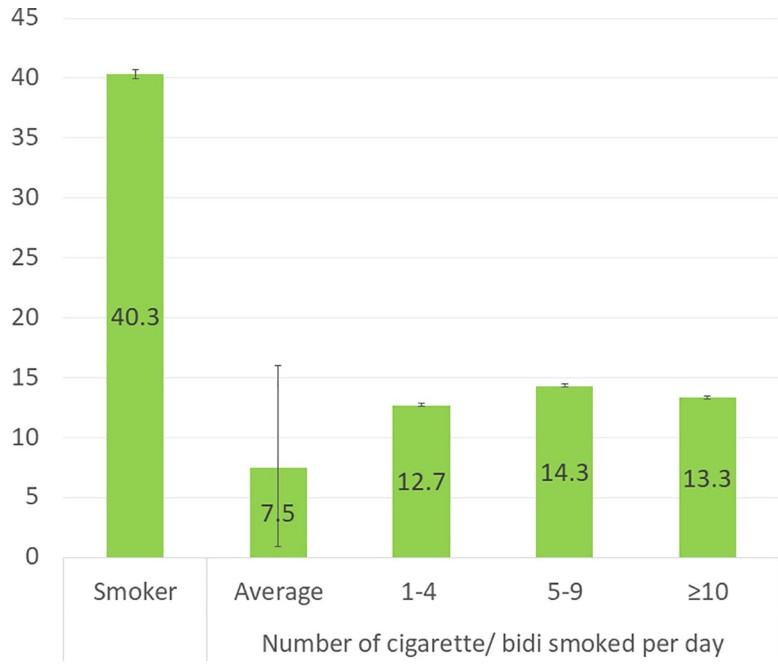

**Fig 1. Prevalence of smoking among youth male, and number of cigarette/bidi smoked per day by the smokers.**

Respondents who believe that smoking brings the risks of chronic diseases and is harmful to other non-smokers are less likely to smoke cigarette/bidi than their counterparts.

Table 2 shows the results of binary logistic regression model that yields the odds of being cigarette/bidi smoker by categories of the explanatory variables. The likelihood of being

**Table 2. Determinants of cigarette/bidi smoking among youth male in rural Mymensingh.**

| Variables | β | SE(β) | Odds ratio | 95% CI |
|---|---|---|---|---|
| Age | 0.28 | 0.07 | 1.33*** | 1.17–1.51 |
| Education | | | | |
| Below secondaryⓇ | | | 1.00 | - |
| Secondary | -0.74 | 0.51 | 0.48 | 0.17–1.30 |
| Higher secondary and above | -0.80 | 0.56 | 0.45 | 0.15–1.34 |
| Occupation | | | | |
| Day labourerⓇ | | | 1.00 | - |
| Service | -1.02 | 0.96 | 0.36 | 0.06–2.39 |
| Business | -1.91 | 0.78 | 0.15* | 0.03–0.68 |
| Students | -0.77 | 0.91 | 0.47 | 0.08–2.74 |
| Unemployed | -1.72 | 0.97 | 0.18 | 0.03–1.20 |
| Monthly income (in BDT) | | | | |
| No incomeⓇ | | | 1.00 | - |
| < 5000 | 2.08 | 0.79 | 7.97** | 1.68–37.74 |
| ≥5000 | 0.95 | 0.87 | 2.60 | 0.47–14.34 |
| Family's monthly income (in BDT) | | | | |
| < 15000Ⓡ | | | 1.00 | - |
| 15000–25000 | -0.96 | 0.41 | 0.38* | 0.17–0.86 |
| > 25000 | -0.30 | 0.33 | 0.75 | 0.39–1.43 |
| Financially dependent on the family | | | | |
| DependentⓇ | | | 1.00 | - |
| Independent | 0.78 | 0.75 | 2.18 | 0.50–9.41 |
| Current smoking status of father | | | | |
| NoⓇ | | | 1.00 | - |
| Yes | 0.92 | 0.30 | 2.51** | 1.39–4.53 |
| Current smoking status of brother | | | | |
| NoⓇ | | | 1.00 | - |
| Yes | 1.06 | 0.37 | 2.88** | 1.39–5.96 |
| Current smoking status of close friend | | | | |
| NoⓇ | | | 1.00 | - |
| Yes | 2.25 | 0.25 | 9.85*** | 5.85–15.27 |
| Smoking brings the risk of chronic disease | | | | |
| NoⓇ | | | 1.00 | - |
| Yes | -2.09 | 1.46 | 0.12 | 0.01–2.18 |
| Smoking is harmful to other non-smokers | | | | |
| NoⓇ | | | 1.00 | |
| Yes | -1.86 | 0.84 | 0.16* | 0.03–0.81 |

**Notes:** β: regression coefficient; BDT: Bangladesh currency–Taka; CI: Confidence interval; Ⓡ: Reference Category; SE: Standard error; Level of significance

***: p<0.001

**: p<0.01

*: p<0.05.

cigarette/bidi smoker increases with the increase in age (Odds ratio: 1.33 and 95% confidence interval: 1.17–1.51). The odds of being cigarette/bidi smoker for business owner is 0.15 (0.03–0.68) times less than day labourer. Compared to respondents with no income, respondents with monthly income (either <BDT 5,000 or ≥BDT 5,000) are more likely to smoke cigarette/bidi. Respondents coming from higher economic status (i.e. family's monthly income ≥BDT 15,000) have lower odds of being cigarette/bidi smokers than those coming from lower economic status (i.e. family's monthly income <BDT 15,000). Respondents whose fathers are cigarette/bidi smokers are more likely to smoke cigarette/bidi than those having non-smoker fathers; also, whose brothers and close friends are cigarette/bidi smokers are more likely to smoke cigarette/bidi than their counterparts. Respondents who believe that smoking is harmful to other non-smokers are significantly less likely to smoke cigarette/bidi than their counterparts.

## Discussion

To the best of our knowledge, this is the first study that reveals the determinants of cigarette/bidi smoking among youth male in rural Mymensingh of Bangladesh. The result of this study exhibits that four in 10 youth are cigarette/bidi smokers in Trishal Upazila of Mymensingh district. The determinants of cigarette/bidi smoking among youth male include age, occupation, monthly income, family's monthly income, cigarette/bidi smoking status of father, brother and close friends, and knowledge about harmfulness of smoking.

Higher age is associated with higher likelihood of cigarette/bidi smoking among youth male in the current study. In previous studies, age is also found as one of the determinants of smoking among the youth [18, 36]. This might be due to the fact that the possibility of being involved in various risky practices, particularly among youth male, increases with the increase in age. Also, in Bangladesh, older youth male have some cash in their hands with which they are capable of buying cigarette/bidi.

Day labourers have the highest odds of smoking cigarette/bidi in the current study. In the United States, blue-collar workers, specifically construction workers in the absence of their workplace rules against smoking, are more likely to be smokers than white-collar workers [37]. Restrictions of smoking at workplaces, mainly smoke-free policies are reported promising for reducing smokers in Canada [38]. In Bangladesh, while service holders, businessmen and students need to comply with smoking restrictions at workplaces/schools, day labourers face almost no smoking restriction at workplaces, and unemployed youth have their limitations in expending money on smoking. No smoking restriction at workplaces could be the possible reason for the highest likelihood of being smokers among day labourers in the study area. Restricting smoking for day labourers at workplaces may reduce prospective smokers among youth male in Bangladesh.

There is an independent positive association of personal income with smoking among adolescents aged 14–17 years in six European countries [39]. Economic inequality has also direct correspondence with tobacco use [40]. Youth male with personal income, compared to those with no income, have higher likelihood of cigarette/bidi smoking in the current study, which is in a line with the previous studies [41, 42]. When youth have extra pocket money, they spend the money for their recreation. Generally, youth male, particularly with their friends, consider smoking as a pastime and therefore expend on smoking cigarette/bidi in Bangladesh. Providing more valuable options, e.g. recreational football game to the youth may be effective in reducing the number of smokers and keeping the youth fit and active.

Consistent with previous literature [43], respondents with lower family income have higher likelihood of smoking cigarette/bidi in our study areas. Paternal smoking, specifically smoking

habit of father was reported to increase the risk of youth smoking [44]. The current study also finds higher likelihood of smoking cigarette/bidi among youth male whose fathers are smokers. In a similar vein, brother's smoking is found to increase the risk of youth smoking in the study areas. Smoking among male is found to be associated with the smoking behaviour of siblings in different settings [45, 46]. These findings indicate that smoking behaviour passes from one generation to another as well as within the same generation. The possible reason might be that the children who grow up by seeing their fathers and brothers smoking, those children may try to make a trail at some later points in time, and may become a regular smoker at their youth stages. Therefore, to prevent smoking among youth male in a family, the older adults– primarily fathers and male siblings of the family should stop smoking in Bangladesh.

Friend smoking is by far the strongest factor to increase the risk of youth smoking in different settings [18, 36, 44, 45, 47, 48]. The current study also exhibits higher likelihood of smoking cigarette/bidi among the respondents whose friends are smokers, compared those with non-smoker friends. Since this study, along with the extant literature, yields the greater peer influences on youth smoking, we suggest families to pay a great attention to their youth members to prevent them from smoking. Similar to the findings of the studies [49, 50], the respondents who believe that smoking is harmful to other non-smokers are less likely to smoke cigarette/bidi in the study areas. Awareness needs to be created through different channels about deadly hazardous effects of smoking and its harmful consequences not only for the smokers and their families but also for the nation as a whole. In this regard, educational institutions and mass media may play an important role.

The current study has some limitations. Being a cross-sectional one, this study does not permit casual association of the explanatory variables with the outcome variable. The data come from rural areas of Trishal Upazila of Mymensingh district, and thus may not be generalizable to entire youth male in Bangladesh. Majority of the respondents (67%) were students, and it limits the generalizability of the current findings among the occupational categories of the youth male in the study area. Despite the limitations, as the pioneer study, it demonstrates a detailed analysis of the various socio-demographic determinants of youth smoking, which is generalizable to rural youth male of Mymensingh district in Bangladesh.

## Conclusion

In sum, as the first study, it provides the determinants of cigarette/bidi smoking among youth male of the rural areas of Mymensingh district. Based on the above discussion, careful planning should be made to prevent smoking among youth, as the youth are the main instruments for future development of the country. Relevant authorities may consider the following five recommendations in formulating strong enforcement measures along with revising or redrawing the existing smoking policies to discourage smoking among the youth. The recommendations are: (i) restricting smoking for day labourers at workplaces, (ii) providing more valuable options to the youth for pastimes, e.g. recreational football game, (iii) families to pay due attention to their youth members to prevent them from smoking, (iv) the older adults of the families– primarily fathers and male siblings stop smoking, and (v) educational institutions and mass media create awareness about the hazardous effects and harmful consequences of smoking not only for the smokers and their families but also for the entire country.

## Supporting information

**S1 File.**
(PDF)

**S2 File.**
(PDF)

**S3 File.**
(SAV)

## Acknowledgments

The authors gratefully acknowledge the Research and Extension Center, Jatiya Kabi Kazi Nazrul Islam University, Bangladesh, for providing the scope to conduct this study.

## Author Contributions

**Conceptualization:** K. M. Mustafizur Rahman, Md. Ismail Tareque.

**Data curation:** K. M. Mustafizur Rahman.

**Formal analysis:** K. M. Mustafizur Rahman.

**Methodology:** K. M. Mustafizur Rahman, Md. Ismail Tareque.

**Supervision:** Md. Ismail Tareque.

**Writing – original draft:** K. M. Mustafizur Rahman.

**Writing – review & editing:** K. M. Mustafizur Rahman, Md. Ismail Tareque.

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
