## [Decision Letter · Decision Letter 0]

8 Oct 2020

PONE-D-20-25908

Determinants of cigarette/ bidi smoking among youth male in rural Mymensingh of Bangladesh: A cross-sectional study

PLOS ONE

Dear Dr. Tareque,

Thank you for submitting your manuscript to PLOS ONE. After careful consideration, we feel that it has merit but does not fully meet PLOS ONE’s publication criteria as it currently stands. Therefore, we invite you to submit a revised version of the manuscript that addresses the points raised during the review process.

We look forward to receiving your revised manuscript.

Kind regards,

Russell Kabir, PhD

Academic Editor

PLOS ONE

Journal Requirements:

2. You indicated that ethical approval was not necessary for your study. We understand that the framework for ethical oversight requirements for studies of this type may differ depending on the setting and we would appreciate some further clarification regarding your research. Could you please provide further details on why your study is exempt from the need for approval and confirmation from your institutional review board or research ethics committee (e.g., in the form of a letter or email correspondence) that ethics review was not necessary for this study? Please include a copy of the correspondence as an "Other" file.

3. In the Methods, please clarify that participants provided oral consent. Please also state in the Methods:

- Why written consent could not be obtained

- Whether the Institutional Review Board (IRB) approved use of oral consent

- How oral consent was documented

For more information, please see our guidelines for human subjects research: " ext-link-type="uri" xlink:type="simple">https://journals.plos.org/plosone/s/submission-guidelines#loc-human-subjects-research"

4. Please provide a sample size and power calculation in the Methods, or discuss the reasons for not performing one before study initiation.

5. To comply with PLOS ONE submission guidelines, in your Methods section, please provide additional information regarding your statistical analyses. For more information on PLOS ONE's expectations for statistical reporting, please see https://journals.plos.org/plosone/s/submission-guidelines.#loc-statistical-reporting.

6. Please include additional information regarding the survey or questionnaire used in the study and ensure that you have provided sufficient details that others could replicate the analyses. For instance, if you developed a questionnaire as part of this study and it is not under a copyright more restrictive than CC-BY, please include a copy, in both the original language and English, as Supporting Information.

7.We note that you have indicated that data from this study are available upon request. PLOS only allows data to be available upon request if there are legal or ethical restrictions on sharing data publicly. For more information on unacceptable data access restrictions, please see http://journals.plos.org/plosone/s/data-availability#loc-unacceptable-data-access-restrictions.

Reviewers' comments:

Reviewer's Responses to Questions

**Comments to the Author**

1. Is the manuscript technically sound, and do the data support the conclusions?

Reviewer #1: Partly

Reviewer #2: Yes

Reviewer #3: Yes

2. Has the statistical analysis been performed appropriately and rigorously? 

Reviewer #1: No

Reviewer #2: Yes

Reviewer #3: Yes

3. Have the authors made all data underlying the findings in their manuscript fully available?

Reviewer #1: Yes

Reviewer #2: Yes

Reviewer #3: Yes

4. Is the manuscript presented in an intelligible fashion and written in standard English?

Reviewer #1: No

Reviewer #2: Yes

Reviewer #3: Yes

5. Review Comments to the Author

Reviewer #1: Determinants of cigarette/ bidi smoking among youth male in rural Mymensingh of Bangladesh: A cross-sectional study

In the above study the authors identified determinants of smoking cigarette/bidi among the

youth male of a rural district of Mymensingh in Bangladesh. My review of the manuscript and recommendations to the authors are enclosed.

Abstract:

1. Data source should be mentioned in the abstract (not just mentioning a secondary source, instead of which secondary source)

Introduction:

2. Line 104-108: the authors describes there is there is a paucity of research to identify the factors responsible for smoking among the youth in Bangladesh which should be rephrased. In fact, there are plenty of research on these.

3. It would be useful to provide more information in the research that has been conducted so far on the population in question.

Methods:

4. The authors included 385 youth data in the study, what were the criterion to enroll an individual in the study. Inclusion exclusion should be explained in detail with reference to the original study protocol.

5. It looks like study sample included only 15-24 years old male individuals. What was the justification on restricting the sample to a specific age group?

6. How the sample size was determined. Consider adding a power analysis to the methods section.

7. What questions were asked to the study participants? Was the questionnaire validated? Questionnaire should be included as a supplemental material for review.

8. How the financial dependency on the family defined/measured? Please consider providing the operational definitions in the methods section

9. There is a very apparent occupation level bias in this sample. Why is that the case? 67% sample are students? Does that measure up with data on the number of smokers in Bangladesh?

Analysis:

10. Did you check the possible collinearity of the independent variables?

11. Why age is considered as categorical? What is the basis of these categories? Why don't consider age a continuous and check if square-Age has significant effect on the outcome?

12. Needs model selection procedure in detail

Results:

13. Table 1 should be comparing the outcome variable by the predictors instead of mentioning the percentages. Which does not provide much information. Table 1 and 2 can be combined.

14. A very important question "number of cigarettes smoked per day” was not included in the analysis.

Discussion

15. Consider updating the discussion providing contrast with studies on similar study setting; not other country o national lavel.

16. An Upazila level sample should not be generalized to national.

Reviewer #2: This is a clearly presented manuscript.

Sentence themes and structure needs to be checked and corrected.

Introduction

Line 61-63: Is this data annual estimates of mortality?

Line 105-108: What is the age range of university students in Bangladesh?

Methods

Line 114 -117: has the data been published before? If yes, please state the reference of the data source. Otherwise, it may not be appropriate as secondary data

Line122-128: This seems to be a multistage sampling technique.

Line 129-132: Give a detailed description of study questionnaire here. How was the questionnaire administered, self or interviewer?

Line 161-164: It is prefered to include all variable in the regression model, rather than using the phased approach to selecting explanatory variables.

Line191: The table does not clearly identify explanatory variables with p0.2 for inclusion in the regresion model.

Results

Table 2 3 needs to redrawn and properly titled for better clarity and understanding by readers.

Discussion

Line 220: Repitition of results in the discussion section should be minimized.

Reviewer #3: Dear editor,

The subject area that authors have studied is important and the manuscript adds valuable information within the field of cigarette smoking. Generally, the manuscript is well presented and informative. In my assessment, the manuscript satisfies the publication criteria of PLOS ONE.

Minor comments and suggestions:

Provide the confidence interval for the prevalence of cigarette smoking.

Kindly use graphs and charts to have a clearer picture of the research findings.

The authors needs to provide sample size determination procedures and assure its representativeness.

6. PLOS authors have the option to publish the peer review history of their article (what does this mean?). If published, this will include your full peer review and any attached files.

Reviewer #1: No

Reviewer #2: No

Reviewer #3: **Yes: **Muluneh Alene Addis

---

## [Author Response · Author response to Decision Letter 0]

2 Dec 2020

Date: December 3, 2020

Professor Russell Kabir, PhD

Academic Editor

PLOS ONE

Subject: Revision and re-submission of manuscript, PONE-D-20-25908

Dear Professor Kabir:

Thank you very much for providing us the opportunity to revise and resubmit the manuscript “Determinants of cigarette/ bidi smoking among youth male in rural Mymensingh of Bangladesh: A cross-sectional study” for publication in PLoS ONE. We would also like to thank the three anonymous reviewers for their valuable comments. Please find below our responses, item-by-item, to the comments.

The current study used data from the project “Knowledge, awareness and practices among youth smokers in Trishal Upazila under Mymensingh district: A micro-survey study’’ funded by the Research and Extension Center, Jatiya Kabi Kazi Nazrul Islam University (JKKNIU), Bangladesh. The data and associated report were supposed to be published by the funder (The Research and Extension Center, JKKNIU). Recently, we happen to know that the funder is not going to publish the data and report due to their internal circumstances. We are denoting the data as primary data this time, in Abstract and Methods sections. The ethics committee of the Research and Extension Center, JKKNIU granted a waiver of ethical approval, as the project had no minimal risk to the subjects. The waiver of ethical approval letter is attached as other file. As there are no ethical or legal restrictions on sharing our data set, we have attached the complete dataset (S3 file) as Supporting information. The questionnaire, in both the original language (Bangla) and English, is also attached as Supporting information (S1 and S2 files).

We have also attached the revised manuscript, incorporating the reviewers’ comments and suggestions, with the edited text highlighted in yellow. We hope that our revisions are acceptable to you. 

Sincerely,

Md. Ismail Tareque, PhD

Professor, Department of Population Science and HRD, University of Rajshahi, Bangladesh

E-mail: tareque_pshd@yahoo.com

tarequemi_pops@ru.ac.bd

 and 

Founding member of the National Young Academy of Bangladesh (NYAB)

NYAB Website: https://nyabangladesh.org/md-ismail-tareque/

Google Scholar: https://scholar.google.com/citations?user=bpcialYAAAAJhl=en

ResearchGate: https://www.researchgate.net/profile/Md_Ismail_Tareque

Personal Website: http://103.79.117.242/ru_profile/public/teacher/22801966/profile 

Journal Requirements:

Response: We have formatted our manuscript according to PLOS ONE's style.

2. You indicated that ethical approval was not necessary for your study. We understand that the framework for ethical oversight requirements for studies of this type may differ depending on the setting and we would appreciate some further clarification regarding your research. Could you please provide further details on why your study is exempt from the need for approval and confirmation from your institutional review board or research ethics committee (e.g., in the form of a letter or email correspondence) that ethics review was not necessary for this study? Please include a copy of the correspondence as an "Other" file.

Response: The current study used data from the project “Knowledge, awareness and practices among youth smokers in Trishal Upazila under Mymensingh district: A micro-survey study’’ funded by the Research and Extension Center, Jatiya Kabi Kazi Nazrul Islam University (JKKNIU), Bangladesh. The executive committee of the Research and Extension Center, JKKNIU examined the project’s proposal, design and questionnaire, and approved a small grant to complete the project. The ethics committee of the Research and Extension Center, JKKNIU granted a waiver of ethical approval, as the project had no minimal risk to the subjects. Nevertheless, publishing an article from this project was one of the terms and conditions. As suggested, the “waiver of ethical approval letter” is attached as other file. 

3. In the Methods, please clarify that participants provided oral consent. Please also state in the Methods:

- Why written consent could not be obtained

- Whether the Institutional Review Board (IRB) approved use of oral consent

- How oral consent was documented

For more information, please see our guidelines for human subjects research: https://journals.plos.org/plosone/s/submission-guidelines#loc-human-subjects-research"

Response: In “Ethical considerations” section under “Methods” section, we have clarified why only verbal consent was obtained rather than written consent. We have added the following:

At the very beginning of the project questionnaire, a consent statement emphasizing voluntary participation in the study on knowledge, awareness and practices about smoking among youth assured that the respondents’ information will be kept strictly confidential and the data will be utilized only for research purpose. The ethics committee of the Research and Extension Center, JKKNIU granted a waiver of ethical approval, as the project had no minimal risk to the subjects. All respondents were asked to provide verbal consent after being read the consent statement. They were not asked for written consent as some of them might have inability in reading and writing, which could make them reluctant to respond to the interview, and consequently the data collection process would have been jeopardized. Any identifying information was removed from the dataset.

4. Please provide a sample size and power calculation in the Methods, or discuss the reasons for not performing one before study initiation.

Response: The minimum required sample size and its calculation procedure is provided in “Data” section under “Methods” section.

5. To comply with PLOS ONE submission guidelines, in your Methods section, please provide additional information regarding your statistical analyses. For more information on PLOS ONE's expectations for statistical reporting, please see https://journals.plos.org/plosone/s/submission-guidelines.#loc-statistical-reporting.

Response: We have adhered to PLOS ONE's expectations for statistical reporting, and updated “Statistical analysis” section. We have provided additional information on multicollinearity check, sampling weights and the name and version of the software package used for data analysis.

6. Please include additional information regarding the survey or questionnaire used in the study and ensure that you have provided sufficient details that others could replicate the analyses. For instance, if you developed a questionnaire as part of this study and it is not under a copyright more restrictive than CC-BY, please include a copy, in both the original language and English, as Supporting Information.

Response: The questionnaire, in both the original language (Bangla) and English, is attached as Supporting information. Please see S1 and S2 files.

7. We note that you have indicated that data from this study are available upon request. PLOS only allows data to be available upon request if there are legal or ethical restrictions on sharing data publicly. For more information on unacceptable data access restrictions, please see http://journals.plos.org/plosone/s/data-availability#loc-unacceptable-data-access-restrictions.

Response: There are no ethical or legal restrictions on sharing our data set. We have attached the complete dataset (S3 file) as Supporting information.

Response: We have attached the complete dataset “S3 File” as Supporting information.

Response: Thank you very much.

Journal Requirements from PLOS ONE: Your submission PONE-D-20-25908R1 - [EMID:1c27b98ff85ce4fe]; Date: November 24, 2020

1) Please ensure that you refer to Figure 1 in your text as, if accepted, production will need this reference to link the reader to the figure.

Response: We refer to Figure 1 in text this time (line# 236).

2) Thank you for your responses to our requests. However, during our internal review, we noted that you provided a document that details the terms of the project funding. At this time, we are requesting that you instead submit an approval or confirmation from your institutional review board or research ethics committee (e.g., in the form of a letter or email correspondence) that ethics review was not necessary for this study. Thank you for your attention to this request.

Response: We have provided “waiver of ethical approval letter” as “other” file in the journal’s submission site.

Journal Requirements from PLOS ONE: PLOS ONE: Your submission PONE-D-20-25908R1 - [EMID:26078aec6d56a1cd]; Date: December 2, 2020

1) Thank for you for stating in your response to reviewer letter 'We refer to Figure 1 in text this time (line# 236).'

Unfortunately reference to this figure is a Legend and instruction where to add the figure. As per PLOS guidelines we do require figures to be cited within your text as if accepted, production will need this reference to link the reader to the figure.

Response: We have cited the figure within the text on line# 233. We are very sorry for not being able to address it earlier. Thank you for your kind consideration.

 

Review Comments to the Author

Reviewer #1: Determinants of cigarette/ bidi smoking among youth male in rural Mymensingh of Bangladesh: A cross-sectional study

In the above study the authors identified determinants of smoking cigarette/bidi among the

youth male of a rural district of Mymensingh in Bangladesh. My review of the manuscript and recommendations to the authors are enclosed.

Response: Thank you very much for your insightful comments and suggestions.

Abstract:

1. Data source should be mentioned in the abstract (not just mentioning a secondary source, instead of which secondary source) 

Response: As suggested, we have mentioned the data source. 

Introduction:

2. Line 104-108: the authors describes there is there is a paucity of research to identify the factors responsible for smoking among the youth in Bangladesh which should be rephrased. In fact, there are plenty of research on these.

Response: We agree that there is a good number of research on the prevalence of and factors associated with smoking in Bangladesh. We have included the existing studies, and updated Introduction section. However, none of the existing studies paid particular focus on the youth section of the population in Bangladesh. We have revised Introduction section accordingly, and added the following:

Although there are studies on the determinants of smoking around the world [15–20], in Bangladesh, extant literature has not paid particular focus on the youth section of the population. For example, Flora and others studied 35,446 individuals aged 18 and 90 years, and found an overall prevalence of smoking as 20.5% during 2001-2003 [21]. Using data from 12,155 men aged 15-59 years, Khan and others found the overall prevalence of smoking as 53.6%, with a higher prevalence in slum (59.8%) than non-slum (46.4%) in 2006 [22]. Sreeramareddy and others used nine south and south-east Asian countries’ data sets from Demographic and Health Surveys, and found the prevalence of smoking as 60.0% among Bangladeshi men of age 15-54 years in 2007 [23]. They also found lower education and poverty to be responsible for smoking among Bangladeshi men. Using the Global Youth Tobacco Survey Bangladesh 2007 data, the prevalence of ever cigarette smoking was reported as 15.8% and the prevalence of smoking as 12.3% among boys who were 13-15 years of age and studying in grades 7-10 [24,25]. Among the university male students aged 18-26 years, the prevalence of cigarette smoking was reported as 49.1% [3]. Due to lack of research on smoking among the youth aged 15-24 years in Bangladesh, we set the aim to identify the determinants of smoking among the youth in Bangladesh.

3. It would be useful to provide more information in the research that has been conducted so far on the population in question.

Response: We have included the existing studies, and updated Introduction section. 

Methods:

4. The authors included 385 youth data in the study, what were the criterion to enroll an individual in the study. Inclusion exclusion should be explained in detail with reference to the original study protocol.

Response: Thank you for raising this issue. There were only two inclusion criteria: (a) individuals of age 15-24 years, and (b) male. Those, who did not meet the inclusion criteria, were neither interviewed nor included in this study. These were mentioned in earlier version of this manuscript, but are explicitly stated this time in “Data” section under “Methods” section.

5. It looks like study sample included only 15-24 years old male individuals. What was the justification on restricting the sample to a specific age group?

Response: We discussed the importance of studying youth smoking in Introduction section. As the study was about the youth male, the sample was restricted to 15-24 years old male individuals. According to United Nations, individuals aged 15-24 years are youth (see line# 129 and ref#26).

6. How the sample size was determined. Consider adding a power analysis to the methods section.

Response: The minimum required sample size and its calculation procedure is provided in “Data” section under “Methods” section.

7. What questions were asked to the study participants? Was the questionnaire validated? Questionnaire should be included as a supplemental material for review.

Response: We did not make any validation of the questionnaire. The questionnaire was prepared in the light of the objectives of the study. The detailed questionnaire, in both the original language (Bangla) and English, is provided as Supporting information. Please see S1 and S2 files.

8. How the financial dependency on the family defined/measured? Please consider providing the operational definitions in the methods section

Response: Provided (lines# 171-172). Thank you very much.

9. There is a very apparent occupation level bias in this sample. Why is that the case? 67% sample are students? Does that measure up with data on the number of smokers in Bangladesh?

Response: Thank you for raising the issue. We agree that this is a limitation and could cause a bias in the study. We mentioned it as a limitation (lines# 321-323) as follows:

Majority of the respondents (67%) were students, and it limits the generalizability of the current findings among the occupational categories of the youth male in the study area.

Analysis:

10. Did you check the possible collinearity of the independent variables?

Response: Yes, we did check multicollinearity of the independent variables. This time, we have explicitly mentioned it in “Statistical analysis” section (lines# 188-191) as follows:

Moreover, multicollinearity in the logistic regression analysis was checked by examining the standard errors for the regression coefficients. A standard error larger than 2.0 indicates numerical problems such as multicollinearity among the explanatory variables [35]. No evidence of multicollinearity was observed.

35. Chan YH. Biostatistics 202: logistic regression analysis. Singapore Medical Journal. 2004;45: 149–153.

11. Why age is considered as categorical? What is the basis of these categories? Why don't consider age a continuous and check if square-Age has significant effect on the outcome?

Response: Thank you very much for your suggestion. We have used single year age for regression analysis (lines# 166-167, Table 2). We have revised Table 2 and text accordingly. 

12. Needs model selection procedure in detail

Response: We have revised “Statistical analysis” section.

Results:

13. Table 1 should be comparing the outcome variable by the predictors instead of mentioning the percentages. Which does not provide much information. Table 1 and 2 can be combined.

Response: As suggested, Tables 1 and 2 are combined.

14. A very important question "number of cigarettes smoked per day” was not included in the analysis.

Response: We have provided the information in Fig 1.

Discussion

15. Consider updating the discussion providing contrast with studies on similar study setting; not other country o national lavel.

Response: Due to lack of studies on similar settings, we had to consider studies from national level and other countries. Thank you for your consideration.

16. An Upazila level sample should not be generalized to national.

Response: We totally agree with you. We stated earlier that “the data come from rural areas of Trishal Upazila of Mymensingh district, and thus may not be generalizable to entire youth male in Bangladesh” (lines# 319-321). You may have missed it. 

Thank you very much. 

 

Reviewer #2: This is a clearly presented manuscript.

Sentence themes and structure needs to be checked and corrected.

Response: Thank you very much for your positive comments and consideration. 

Introduction

Line 61-63: Is this data annual estimates of mortality?

Response: Yes, it is. We have revised the sentence (lines# 61-63).

Line 105-108: What is the age range of university students in Bangladesh?

Response: The age range of university students was 18-26 years. We have revised the sentence (lines# 116-117). 

Methods

Line 114 -117: has the data been published before? If yes, please state the reference of the data source. Otherwise, it may not be appropriate as secondary data

Response: Thank you very much for raising the issue. The data and associated report were supposed to be published by the funder (The Research and Extension Center, Jatiya Kabi Kazi Nazrul Islam University, Bangladesh). Recently, we happen to know that the funder is not going to publish the data and report due to their internal circumstances. 

We are now denoting the data as primary data (lines# 35 and 124). 

Line122-128: This seems to be a multistage sampling technique.

Response: Revised as suggested (lines# 137-138).

Line 129-132: Give a detailed description of study questionnaire here. How was the questionnaire administered, self or interviewer?

Response: The questionnaire, in both the original language (Bangla) and English, is attached as Supporting information. Please see S1 and S2 files. On lines 146-150, we have provided a brief discussion of the questionnaire, which was administered by interviewers. We have provided the following:

A well-structured Bangla questionnaire containing demographic (age, number of family members etc.), socio-economic (education, occupation, monthly income etc.) and smoking related (smoking status, duration of smoking, smoking status of family members and friends etc.) questions was administered face-to-face to 385 youth males by well-trained interviewers.

Line 161-164: It is prefered to include all variable in the regression model, rather than using the phased approach to selecting explanatory variables.

Response: Previous studies find that the variables significant at 20% or more level in chi-square tests do not persist in or contribute to regression analysis. We have revised “Statistical analysis” section accordingly, with appropriate references. In sensitivity analysis (results not shown), we considered all variables in a regression model, and found insignificant effect of the variables which were significant at 20% or more level in chi-square tests. Thank you very much for your consideration.

Line191: The table does not clearly identify explanatory variables with p0.2 for inclusion in the regression model.

Response: Thank you for highlighting the issue. We have used boldface to denote the p-values 0.20 (See Table 1 and its note).

Results

Table 2 3 needs to redrawn and properly titled for better clarity and understanding by readers.

Response: Previous Tables 1 and 2 are combined (Currently Table 1). Previous Table 3 becomes Table 2. We have revised the titles as well.

Discussion

Line 220: Repitition of results in the discussion section should be minimized.

Response: Revised as suggested. 

Thank you very much.

Reviewer #3: Dear editor,

The subject area that authors have studied is important and the manuscript adds valuable information within the field of cigarette smoking. Generally, the manuscript is well presented and informative. In my assessment, the manuscript satisfies the publication criteria of PLOS ONE.

Response: Thank you very much for your positive comments and suggestions.

Minor comments and suggestions:

Provide the confidence interval for the prevalence of cigarette smoking.

Response: As suggested, we have provided the confidence interval.

Kindly use graphs and charts to have a clearer picture of the research findings.

Response: Thank you very much for raising the issue. We have taken your point, and included a graph for highlighting the prevalence of smoking and number of cigarette/ bidi smoked per day (Fig 1). 

The authors needs to provide sample size determination procedures and assure its representativeness.

Response: Provided in “Data” section as suggested.

Thank you very much.

---

## [Editor Report · Decision Letter 1]

8 Dec 2020

Determinants of cigarette/ bidi smoking among youth male in rural Mymensingh of Bangladesh: A cross-sectional study

PONE-D-20-25908R1

Dear Dr. Tareque,

We’re pleased to inform you that your manuscript has been judged scientifically suitable for publication and will be formally accepted for publication once it meets all outstanding technical requirements.

Kind regards,

Russell Kabir, PhD

Academic Editor

PLOS ONE
---

## [Editor Report · Acceptance letter]

11 Dec 2020

PONE-D-20-25908R1 

Determinants of cigarette/ bidi smoking among youth male in rural Mymensingh of Bangladesh: A cross-sectional study 

Dear Dr. Tareque:

I'm pleased to inform you that your manuscript has been deemed suitable for publication in PLOS ONE. Congratulations! Your manuscript is now with our production department. 

Kind regards, 

on behalf of

Dr. Russell Kabir 

Academic Editor

PLOS ONE